# Rtf1 HMD domain facilitates global histone H2B monoubiquitination and regulates morphogenesis and virulence in the meningitis-causing pathogen *Cryptococcus neoformans*

**Yixuan Jiang[1†], Ying Liang[1†], Fujie Zhao[1], Zhenguo Lu[1], Siyu Wang[1], Yao Meng[1], Zhanxiang Liu[1], Jing Zhang[1]\*, Youbao Zhao[1,2,3]\***

[1]College of Veterinary Medicine, Henan Agricultural University, Zhengzhou, China; [2]Key Laboratory of Quality and Safety Control of Poultry Products, Ministry of Agriculture and Rural Affairs, Zhengzhou, China; [3]Henan Province Key Laboratory of Animal Food Pathogens Surveillance, Zhengzhou, China

**\*For correspondence:**
zhangjing@henau.edu.cn (JZ);
zhaoyoubao@henau.edu.cn (YZ)

[†]These authors contributed equally to this work

**Competing interest:** The authors declare that no competing interests exist.

## eLife Assessment

This is an **important** study that connects the polymerase-associated factor 1 complex (Paf1C) with Histone 2B monoubiquitination and the expression of genes key to virulence in Cryptococcus neoformans. The provided information is **convincing** and has the potential to open several opportunities to further understand the basic biology of this significant human fungal pathogen.

**Abstract** Rtf1 is generally considered to be a subunit of the Paf1 complex (Paf1C), which is a multifunctional protein complex involved in histone modification and RNA biosynthesis at multiple stages. Rtf1 is stably associated with the Paf1C in *Saccharomyces cerevisiae*, but not in other species including humans. Little is known about its function in human fungal pathogens. Here, we show that Rtf1 is required for facilitating H2B monoubiquitination (H2Bub1), and regulates fungal morphogenesis and pathogenicity in the meningitis-causing fungal pathogen *Cryptococcus neoformans*. Rtf1 is not tightly associated with the Paf1C, and its histone modification domain (HMD) is sufficient to promote H2Bub1 and the expression of genes related to fungal mating and filamentation. Moreover, Rtf1 HMD fully restores fungal morphogenesis and pathogenicity; however, it fails to restore defects of thermal tolerance and melanin production in the *rtf1Δ* strain background. The present study establishes a role for cryptococcal Rtf1 as a Paf1C-independent regulator in regulating fungal morphogenesis and pathogenicity, and highlights the function of HMD in facilitating global H2Bub1 in *C. neoformans*.

## Introduction

In eukaryotes, gene transcription is regulated by dynamic changes in chromatin. The posttranslational modifications of core histones, including acetylation, methylation, and ubiquitination, represent major mechanisms by which cells alter the chromatin structural properties and regulate gene transcription (*Taylor and Young, 2021*; *Yun et al., 2011*). Among them, the monoubiquitination of a lysine (K) residue on the C-terminal of histone H2B (H2Bub1) is a conserved modification that occurs on H2B

K120 residue in *Homo sapiens* and K123 residue in *Saccharomyces cerevisiae* (*Fetian et al., 2023*; *Piro et al., 2012*). H2Bub1 is enriched at regions of active transcription but plays roles in both gene activation and repression (*Batta et al., 2011*; *Sun and Allis, 2002*; *Briggs et al., 2002*). In addition, H2Bub1 is required for di- and trimethylation of H3 K4 and H3 K79, subsequently modulating chromatin accessibility (*Worden and Wolberger, 2019*; *Dover et al., 2002*; *Kim and Buratowski, 2009*; *Kim et al., 2012*; *Kim et al., 2009*).

The ubiquitin conjugase (E2) Rad6 and the ubiquitin ligase (E3) Bre1 are responsible for H2Bub1 in *S. cerevisiae* (*Hwang et al., 2003*; *Robzyk et al., 2000*; *Wood et al., 2003a*). In addition, H2Bub1 is regulated by additional factors in yeast and other eukaryotes, among which the conserved polymerase-associated factor 1 (Paf1) complex (Paf1C) is the prominent one (*Piro et al., 2012*; *Krogan et al., 2003*; *Ng et al., 2003*; *Van Oss et al., 2016*; *Wood et al., 2003b*) Paf1C is a multifunctional protein complex, which impacts RNA synthesis at multiple stages (*Francette et al., 2021*; *Jaehning, 2010*; *Squazzo et al., 2002*; *Kim et al., 2010*; *Chen et al., 2009*; *Mueller et al., 2004*; *Penheiter et al., 2005*; *Sheldon et al., 2005*; *Tomson et al., 2011*). Paf1C consists of the subunits Paf1, Ctr9, Cdc73, Rtf1, and Leo1, and the five subunits are stably associated within the complex in *S. cerevisiae* (*Squazzo et al., 2002*; *Koch et al., 1999*; *Mueller and Jaehning, 2002*; *Costa and Arndt, 2000*). In contrast, Rtf1 appears not to be stably associated with Paf1C in human cells, despite the Paf1C is structurally and functionally conserved (*Rozenblatt-Rosen et al., 2009*; *Chu et al., 2013*; *Zhu et al., 2005*). Interestingly, the histone modification domain (HMD) within Rtf1 is both necessary and sufficient for stimulating H2Bub1 in yeast (*Piro et al., 2012*; *Van Oss et al., 2016*). Expression of the Rtf1 HMD alone restores H2Bub1 levels in *S. cerevisiae* mutants deleted for the *RTF1* gene or all five Paf1C subunit-encoding genes (*Fetian et al., 2023*; *Piro et al., 2012*; *Van Oss et al., 2016*). These studies show that Rtf1 is the only Paf1C subunit that is strictly required for deposition of H2Bub1 in vivo. However, little is known about its role in human fungal pathogens.

*Cryptococcus neoformans*, the top-ranked fungus in the WHO fungal pathogen priority list, is a globally distributed opportunistic fungal pathogen that can cause life-threatening cryptococcosis (*Zhao et al., 2023*; *May et al., 2016*). The mortality rate of cryptococcosis is alarmingly high, especially in patients with HIV infection, in whom it ranges from 41% to 61% (*Rajasingham et al., 2022*; *Iyer et al., 2021*). *C. neoformans* can be classified into two serotypes: the serotype A *C. neoformans* and the serotype D *C. deneoformans*. Both *C. neoformans* and *C. deneoformans* undergo yeast-to-hypha transition under inducing conditions, which has been shown to be associated with fungal virulence (*Zhao, 2019*; *Lin and Heitman, 2006*). Thus, deciphering the regulatory mechanisms on fungal morphogenesis and pathogenesis in *C. neoformans* is critical for comprehensive understanding of the nature of pathogen and combating against cryptococcal infection.

In our previous study, we characterized the subunits of complex associated with Set1 (COMPASS) and found that COMPASS-mediated H3K4 methylation (H3K4me) affects yeast-to-hypha transition and virulence in both *C. neoformans* and *C. deneoformans* (*Liu et al., 2023*). We also preliminarily showed that H2Bub1 is required for COMPASS-mediated H3K4me by deletion of *RAD6* and *RTF1* in *C. neoformans* and *C. deneoformans* (*Liu et al., 2023*). However, we set out to characterize the roles of Rtf1 in facilitating global H2Bub1 and to gain comprehensive insights into the epigenetic regulation on fungal morphogenesis and pathogenesis in the human fungal pathogen *C. neoformans*.

## Results

### Rtf1-mediated global H2Bub1 regulates cryptococcal yeast-to-hypha transition

PAF1C subunit Rtf1 functions at the interface between Paf1C and Rad6/Bre1, and is required for deposition of H2Bub1 in all the eukaryotic species examined (*Francette et al., 2021*). We showed that Rtf1 is also required for H2Bub1 and subsequent COMPASS-mediated H3K4me in the *C. deneoformans* reference strain XL280α background (*Figure 1A, B*; *Liu et al., 2023*). Interestingly, loss of H2Bub1 through deleting *RTF1* blocked unisexual yeast-to-hypha transition in *C. deneoformans* (*Figure 1—figure supplement 1A*; *Liu et al., 2023*). To establish the role of Rtf1 in regulating cryptococcal filamentation during bisexual mating, we obtained *RTF1* deletion mutant in the *C. deneoformans* reference strain XL280**a** background through spore dissection from cross between *rtf1Δα* and XL280**a**, and conducted bisexual cross assay under mating-inducing condition on V8 media. The

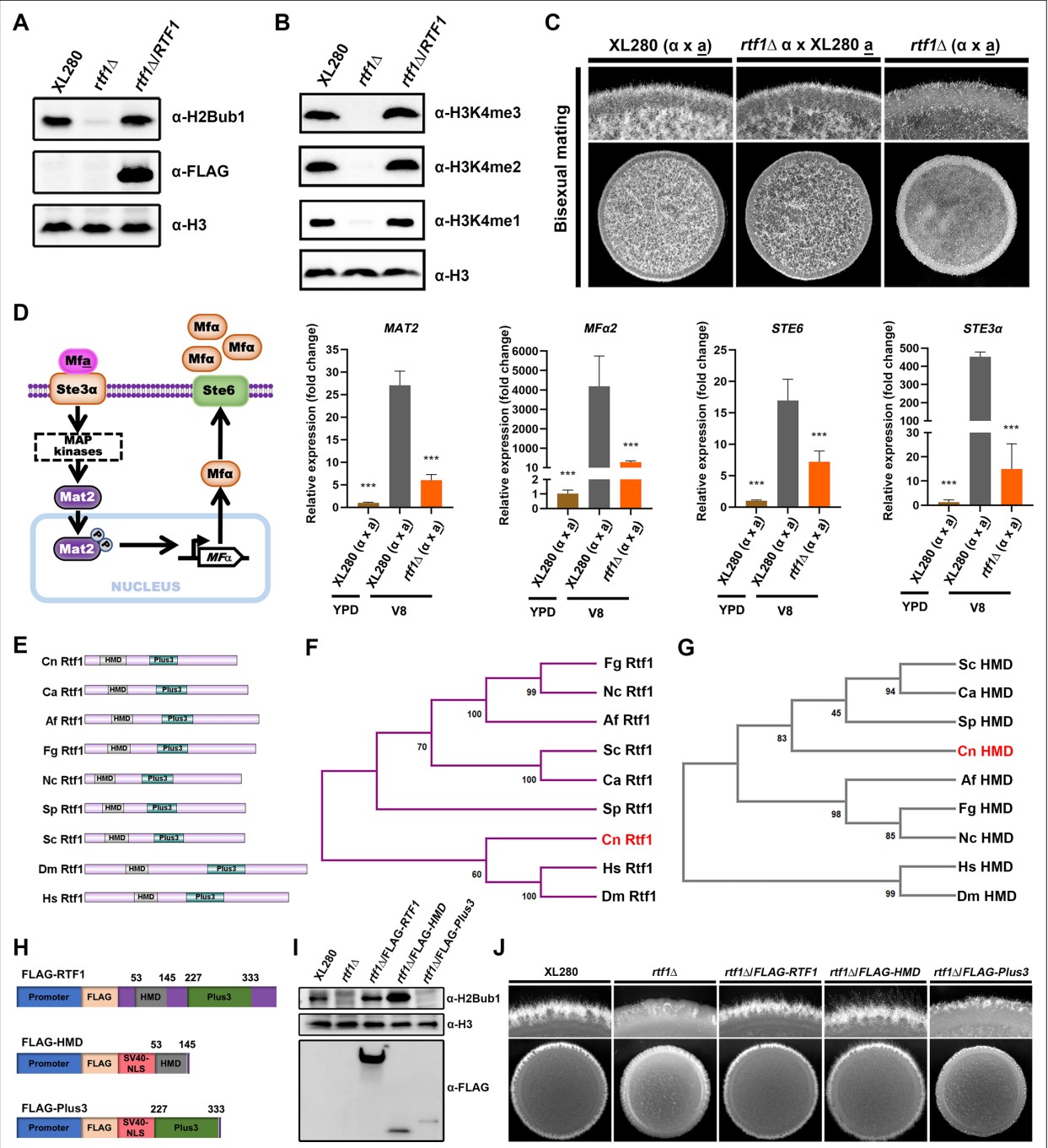

**Figure 1.** Rtf1 regulates cryptococcal bisexual mating by facilitating H2Bub1 via the histone modification domain (HMD). (**A**) Immunoblot analysis of H2Bub1 in *C. deneoformans* wild-type XL280, *rtf1Δ*, and *rtf1Δ/RTF1* strains. (**B**) Immunoblot analysis of H3K4me (including H3K4me1, H3K4me2, and H3K4me3) in *C. deneoformans* wild-type XL280, *rtf1Δ*, and *rtf1Δ/RTF1* strains. (**C**) Colony morphology of cells during bisexual mating between indicated strains. The same volume of cells with opposite mating type at $OD_{600}$ = 3 were mixed, and 3 µl of mixtures were spotted and cultured on V8 media for 2 days at room temperature in dark. (**D**) Schematic diagram of pheromone-dependent signaling pathway and transcript levels of genes involved in pheromone signaling. The mating cells were prepared and cultured on V8 media following the same protocol as the colony morphology assay. After 24 hr, cells were collected for total RNA extraction and qPCR. Bars show the mean ± SD of three biological replicates. Student's *t*-test were used to compare differences between groups. ***, *p* < 0.001.(**E**) Domain structure of Rtf1 homologs in indicated eukaryotes. Cn, *C. deneoformans*; Ca, *Candida albicans*; Af, *Aspergillus fumigatus*; Fg, *Fusarium graminearum*; Nc, *Neurospora crassa*; Sp, *Schizosaccharomyces pombe*; Sc, *S. cerevisiae*; Dm, *Drosophila melanogaster*; Hs, *Homo sapiens*. (**F, G**) Neighbor-joining tree of Rtf1 homologs and their corresponding HMD in indicated eukaryotes. (**H**) Schematic diagram of overexpressing constructs of Rtf1, HMD domain, and Plus3 domain. The constitutive promoter of *TEF1* gene was used to drive

*Figure 1 continued on next page*

*Figure 1 continued*

gene expression. (**I**) Immunoblot analysis of H2Bub1 in strains expressing the indicated proteins. (**J**) Unisexual hyphal formation of indicated strains on V8 media.

The online version of this article includes the following source data and figure supplement(s) for figure 1:

**Source data 1.** Original files for the western blot analysis displayed in *Figure 1A, B, I*.

**Source data 2.** PPT files of western blots for *Figure 1A, B, I*, indicating the relevant bands.

**Figure supplement 1.** The regulation of Rtf1 on filametation and phylogenetic analysis of Plus3 domain in eukaryotes.

mating hyphae during unilateral mating between *rtf1Δα* and XL280**a** were produced at a slightly reduced level compared to mating between reference partners XL280α and XL280**a**, while filamentation was significantly reduced during bilateral mating between *rtf1Δα* and *rtf1Δ**a*** (*Figure 1C*).

During bisexual mating in *C. deneoformans*, mating pheromone (MF) is produced in cells and secreted through the transporter Ste6 (*Hsueh and Shen, 2005*). Secreted pheromone induces mating response by binding to the compatible receptor on the cell surface of opposite mating type (Mf**a** to Ste3α or Mfα to Ste3**a**) (*Shen et al., 2002*; *Wang and Heitman, 1999*). In addition, Mat2, which is a direct downstream transcription factor of the Cpk1 MAPK pathway, regulates the transcription of genes encoding the above-mentioned pheromone sensing proteins (*Lin et al., 2010*; *Figure 1D*). Given the bisexual mating hyphae reduction caused by *RTF1* deletion, we further investigated the effects of *RTF1* deletion on genes involved in bisexual mating at transcript level via qPCR. In comparison to the mating-suppressing condition (YPD media), the transcript levels of *MAT2*, *MFα2*, *STE6*, and *STE3α* were all highly induced under mating-inducing condition (V8 media). However, these inductions were significantly impaired by deletion of *RTF1* (*Figure 1D*). These results strongly indicated that Rtf1 facilitates H2Bub1 and regulates the expression of genes involved in fungal morphogenesis in *C. deneoformans*.

## Ectopic expression of HMD restores global H2Bub1 levels and cryptococcal yeast-to-hypha transition

As the key subunit of Paf1C in mediating histone H2Bub1, Rtf1 is conserved across eukaryotes and consists of two conserved domains, an HMD and a domain that contains three highly conserved positively charged residues (Plus3) (*Figure 1E*). It is worth noting that Rtf1 protein and Plus3 domain in *C. neoformans* is evolutionarily close to higher eukaryotes, such as *H. sapiens* and *Drosophila melanogaster* (*Figure 1F*, *Figure 1—figure supplement 1B, C*), while the HMD domain is distant from higher eukaryotes (*Figure 1G*). To further dissect the roles of Rtf1 HMD and Plus3 in facilitating histone H2Bub1 in *C. deneoformans*, we constructed the truncated versions of Rtf1 that encode HMD (53-145) or Plus3 (227-333) with a nuclear localization sequence (NLS) added in their N terminus, respectively, driven by the constitutive promoter and tagged with FLAG (*Figure 1H*). Interestingly, overexpression of HMD domain itself significantly promoted H2Bub1 to an even higher level in the *rtf1Δ* strain, compared to that in WT strain and the strain overexpressing the full length of *RTF1* (*Figure 1I*), while overexpression of the Plus3 failed to restore H2Bub1 (*Figure 1I*). These results demonstrated that HMD alone is sufficient to facilitate the global H2Bub1 level in *C. deneoformans*.

Our previous studies have demonstrated that H2Bub1 is positively related to the filamentation in *C. neoformans* (*Liu et al., 2023*). Consistently, overexpressing either the full length of *RTF1* or the HMD domain, but not Plus3, promoted the filamentation in *rtf1Δ* strain (*Figure 1J*). To gain an overview of effects on gene expression patterns by the overexpression of HMD domain, we conducted transcriptome profiling by RNA-seq under filamentation-inducing condition (on V8 media). The results showed that the expression levels of 668 genes were significantly changed due to the *RTF1* deletion compared to the WT on V8 media ($|log_2FC| > 1$, adjusted p-value <0.05), with 308 genes significantly upregulated and 360 genes downregulated (*Figure 2A*, *Figure 2—source data 1*). It is worth noting that the downregulated genes are mainly enriched in GO categories related to sexual reproduction, pheromone-dependent signaling, and filamentous growth (*Figure 2—figure supplement 1*). Strikingly, overexpression of HMD domain alone in *rtf1Δ* strain successfully restored the expression of these genes to similar levels as those in wild-type XL280 strain, while overexpression of Plus3 domain failed to do so (*Figure 2A*). In particular, the expression levels of marker genes of filamentous growth (*ZNF2* and *CFL1*) (*Wang et al., 2013*; *Lin et al., 2010*) and genes involved in sexual reproduction and

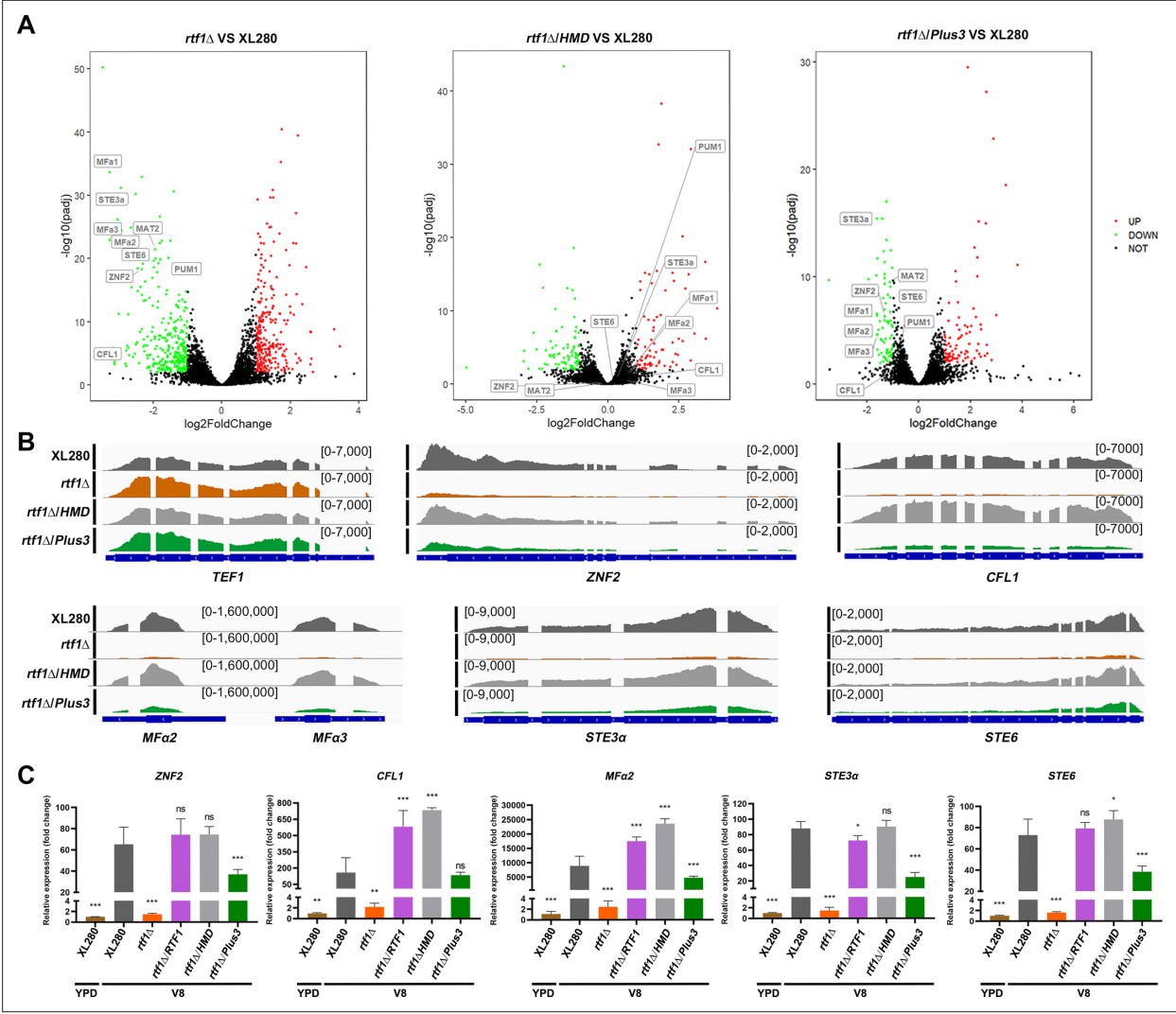

**Figure 2.** The expression of histone modification domain (HMD) alone rescues the downregulation of genes involved in pheromone signaling and filamentous growth due to the deletion of *RTF1*. (**A**) Volcano plots of differentially expressed genes in *rtf1Δ*, *rtf1Δ*/HMD, and *rtf1Δ*/Plus3 relative to wild-type XL280, respectively, during unisexual mating on V8 media. Genes involved in pheromone signaling and filamentous growth were indicated. (**B**) Reads coverage of indicated gene loci in XL280, *rtf1Δ*, *rtf1Δ*/HMD, and *rtf1Δ*/Plus3 strains. Reads coverage at *TEF1* locus served as control. (**C**) qPCR quantification of transcript levels of indicated genes in XL280, *rtf1Δ*, *rtf1Δ*/HMD, and *rtf1Δ*/Plus3 strains cultured on V8. The transcript level of indicated genes were relative to its transcript level in XL280 cells cultured in YPD. Bars show the mean ± SD of three biological replicates. Student's *t*-test were used to compare differences between groups. *, $p < 0.05$; **, $p < 0.01$; ***, $p < 0.001$.

The online version of this article includes the following source data and figure supplement(s) for figure 2:

**Source data 1.** List of differentially expressed genes in *rtf1Δ*, *rtf1Δ*/HMD, and *rtf1Δ*/Plus3 relative to the wild-type XL280 strain on V8.

**Figure supplement 1.** Overexpression of Plus3 domain alone failed to rescue the downregulation of genes enriched in sexual reproduction, pheromone signaling, and filamentous growth due to *RTF1* deletion.

**Figure supplement 2.** Histone modification domain (HMD) alone successfully restores the expression of *MAT2* and *MFα1* in *rtf1Δ* strain.

pheromone-dependent signaling (*MFα*, *STE3α*, and *STE6*) as shown in *Figure 1D* were restored by overexpressing HMD domain alone in the rtf1Δ background (*Figure 2B*, *Figure 2—figure supplement 2*). These findings from transcriptome analyses were further confirmed by qPCR (*Figure 2C*). In addition, the downregulated genes in *rtf1Δ*/Plus3 cells were significantly enriched in common GO categories as the downregulated genes in *rtf1Δ* cells (*Figure 2—figure supplement 1*), relative to the wild-type XL280 strain. Together, these results strongly suggested that HMD domain is sufficient to facilitate global H2Bub1 to promote expression of genes associated with filamentation.

## HMD is sufficient to facilitate global H2Bub1 and the consequent yeast-to-hypha transition

The full-length Rtf1 or HMD domain should properly translocate into the nucleus to facilitate histone H2Bub1. To confirm the function of Rtf1 and HMD domain in facilitating H2Bub1, we artificially intervened their sub-cellular localizations and evaluated the effects of non-nuclear (cell membrane) and

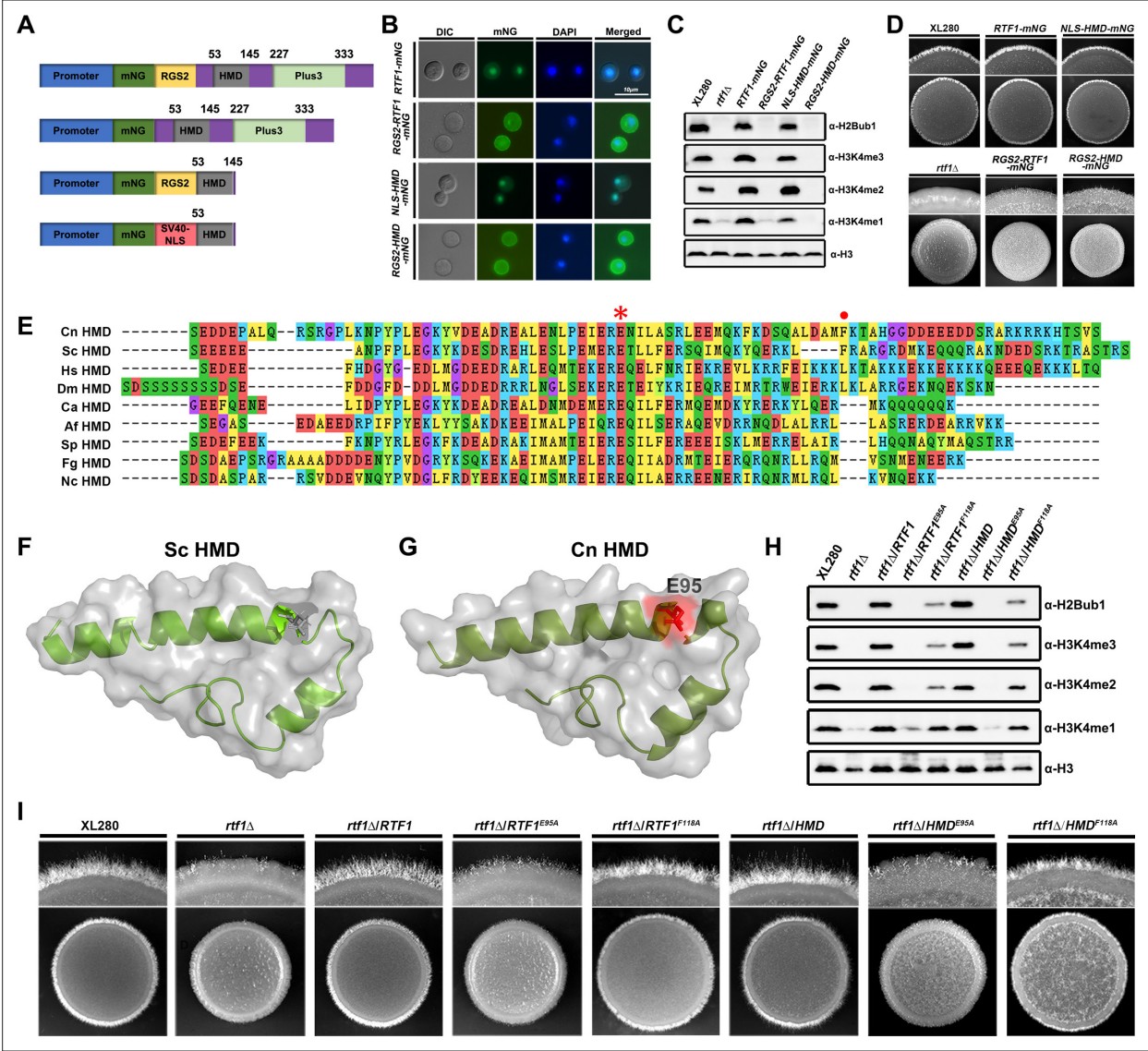

**Figure 3.** The histone modification domain (HMD) alone is sufficient to facilitate global H2Bub1 and restore hyphal formation in *rtf1Δ* strain. (**A**) Schematic diagram of constructs expressing mNG-labeled Rtf1 or HMD domain with cell membrane tag (RGS2) or nuclear localization sequence (NLS). (**B**) Fluorescence analysis of sub-cellular localizations of Rtf1 and HMD with RGS2 and NLS. (**C**) Immunoblot analysis of H2Bub1 and H3K4me in strains expressing the indicated proteins. (**D**) Colony morphology of indicated strains during unisexual mating on V8. (**E**) ClustalW multiple amino acid sequence alignment of the HMD domain in the indicated eukaryotes. The E95 and F118 residues in *C. deneoformans* were indicated with asterisk and dot, respectively. Cn, *C. deneoformans*; Ca, *Candida albicans*; Af, *Aspergillus fumigatus*; Fg, *Fusarium graminearum*; Nc, *Neurospora crassa*; Sp, *Schizosaccharomyces pombe*; Sc, *S. cerevisiae*; Dm, *Drosophila melanogaster*; Hs, *Homo sapiens*. (**F, G**) The 3D structure of *C. deneoformans* HMD domain predicted by SWISS-MODEL with the 3D structure of *S. cerevisiae* HMD domain (5emx) as the template. The conserved E95 residue was indicated in red. (**H**) Immunoblot analysis of H2Bub1 and H3K4me in strains expressing the indicated proteins. (**I**) Colony morphology of indicated strains during unisexual mating on V8.

The online version of this article includes the following source data for figure 3:

**Source data 1.** Original files for the western blot analysis displayed in *Figure 3C, H*.

**Source data 2.** PPT files of western blots for *Figure 3C, H*, indicating the relevant bands.

nuclear localizations on H2Bub1, H3K43me, and filamentation. To achieve cell membrane localization, we fused the full-length Rtf1 and HMD with a cell membrane RGS2-mNeonGreen tag (*Heximer et al., 2001*; *Chen et al., 2020*; *Figure 3A*), and introduced the constructs into the *rtf1Δ* strain, respectively. As indicated by the mNeonGreen fluorescence, HMD domain and the originally nuclear-localized full-length Rtf1 translocated to cell membrane after fusing with the RGS2-mNeongreen tag (*Figure 3B*). Both nuclear-localized Rtf1 and HMD domain restored the levels of H2Bub1, H3K4me, and filamentation (*Figure 3C, D*). In contrast, the non-nuclear-localized full-length Rtf1 or HMD domain failed to restore the levels of H2Bub1, H3K4me, or filamentation in the *rtf1Δ* strain (*Figure 3C, D*). These results further supported the role of Rtf1 HMD domain in facilitating H2Bub1.

Rtf1 HMD domain is conserved from various eukaryotic species, and the residue of glutamine at position 95 (E95, *Figure 3E–G*) has been shown to be critical for the function of Rtf1 (*Tomson et al., 2011*). It is noteworthy that the residue of phenylalanine at position 118 (F118) in *C. neoformans* is as conserved as the residue in *S. cerevisiae* (*Figure 3E*), which is critical for H2Bub1 in yeast, although it is not conserved in other eukaryotic species (*Tomson et al., 2011*). To investigate their roles in cryptococcal Rtf1 HMD domain, we constructed site-mutated alleles of full-length Rtf1 (Rtf1[E95A] and Rtf1[F118A]) and HMD domain (HMD[E95A] and HMD[F118A]) and introduced them into the *rtf1Δ* strain, respectively. Both Rtf1[E95A] and HMD[E95A] failed to restored H2Bub1 and H3K4me levels in the *rtf1Δ* strain, while Rtf1[F118A] and HMD[F118A] partially restored H2Bub1 and H3K4me levels (*Figure 3H*). In consistent with the histone modification outputs, the mutants expressing Rtf1[E95A] or HMD[E95A] showed non-filamentous phenotypes similar as the staring *rtf1Δ* strain, while mutants expressing Rtf1[F118A] or HMD[F118A] produced more filaments than the starting *rtf1Δ* strain (*Figure 3I*). Together, these results demonstrated that Rtf1 HMD domain itself is sufficient to facilitate H2Bub1 and consequent cryptococcal filamentation with E95 as a critical conserved residue.

## Roles of the global H2Bub1 level in cryptococcal virulence factor production

To investigate the role of HMD-mediated H2Bub1 in cryptococcal virulence, we constructed *RTF1* deletion strain and mutants overexpressing the full-length Rtf1, HMD domain, Plus3 domain, or mutated alleles of Rtf1[E95A] and HMD[E95A], respectively, in the clinically isolated serotype A *C. neoformans* H99 strain background. Consistent with what we observed in the serotype D *C. deneoformans*, deletion of *RTF1* abolished H2Bub1 and H3K4me, and overexpressing the full length of Rtf1 and HMD domain alone, but not the Plus3 domain, Rtf1[E95A] or HMD[E95A], successfully restored H2Bub1 and H3K4me (*Figure 4A*). Next, we investigated whether Rtf1 HMD domain is involved in the production of major virulence factors in vitro and pathogenicity in murine models of cryptococcosis. As shown in *Figure 4B*, the *rtf1Δ* mutant had severe growth defect at 39°C, and overexpression of the full-length Rtf1, but not the Plus3 domain, Rtf1[E95A] or HMD[E95A], partially restored the thermal sensitivity of the *rtf1Δ* mutant (*Figure 4B*). Interestingly, overexpression of HMD domain alone restored the growth defect of *rtf1Δ* mutant at 39°C to a level that was worse than the expression of the full length of Rtf1 (*Figure 4B*). Furthermore, the *rtf1Δ* mutant was incapable to produce melanin, and only the full length of Rtf1 restored its melanin production, while the HMD domain alone failed to do so (*Figure 4B*). These results strongly indicate that the full length of Rtf1, but not only the levels of global H2Bub1, is required to regulate thermal tolerance and melanin production in *C. neoformans*.

Capsule production and cell size are known factors that are tightly associated with cryptococcal virulence. We tested the capsule production in *RTF1*-related mutants on the fetal bovine serum (FBS) solid media cultured at 37°C with 5% $CO_2$. The control mutant strain *nrg1Δ* produced no capsule under this condition, and the *rtf1Δ* mutant produced slightly less capsule compared to the wild-type H99 strain in terms of capsule thickness (*Figure 4C*). Interestingly, the cell size of the *rtf1Δ* mutant was significantly larger than the H99 strain under this condition (*Figure 4C, D*), and only overexpression of the full length of Rft1 restored the cell size enlargement phenotype (*Figure 4D*). Consistently, the ratio between diameters of capsule layer and cell body in *rtf1Δ* was significantly smaller than the ratio in the H99 strain (*Figure 4E*). In addition, overexpression of the full-length Rtf1 partially restored the ratio to the level as in the H99 strain, and the HMD, Plus3, Rtf1[E95A], or HMD[E95A] failed to do so (*Figure 4E*). These results showed that the global distribution of H2Bub1 across the chromosome play critical roles in regulating capsule production and cell size in *C. neoformans*.

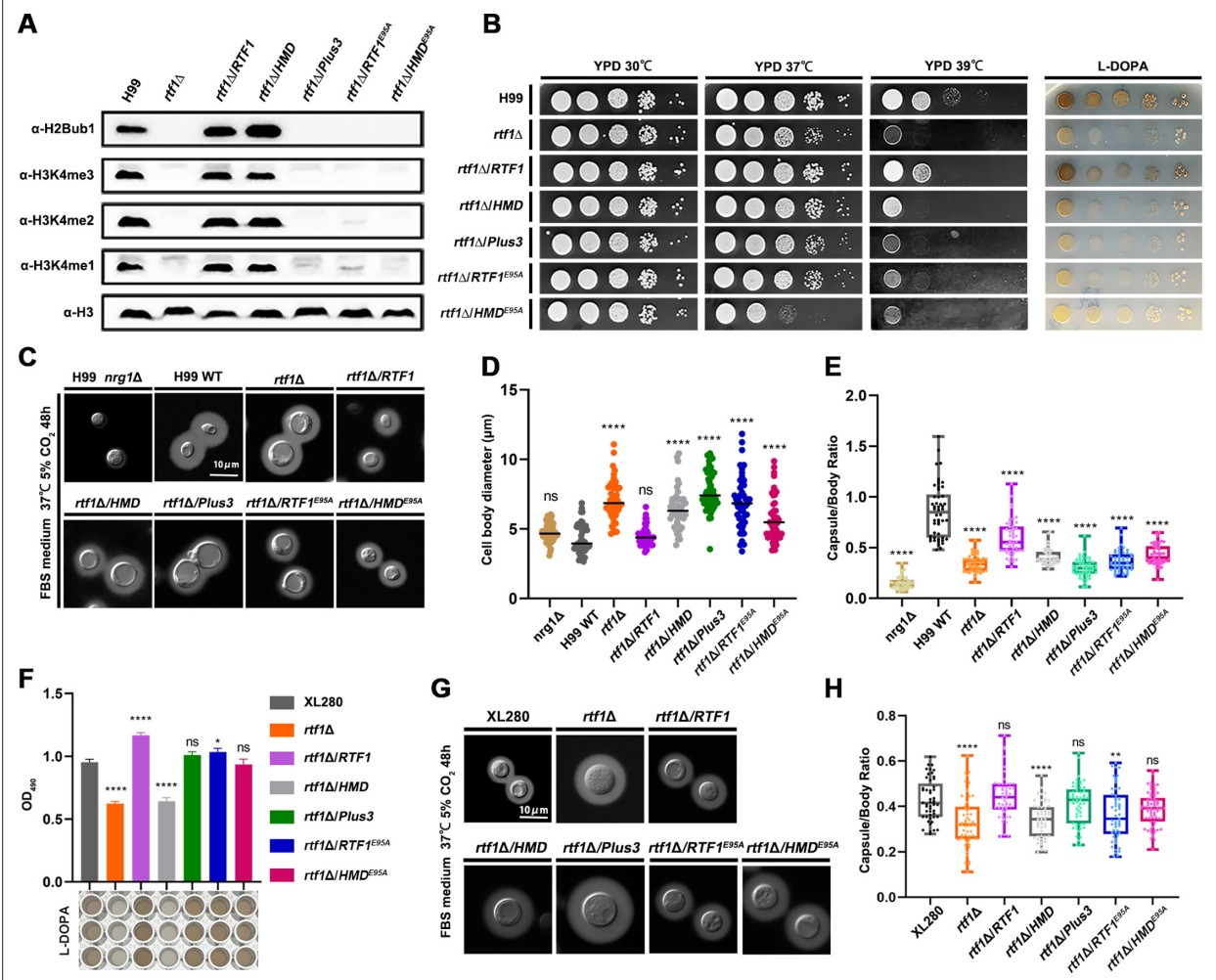

**Figure 4.** Rtf1 and histone modification domain (HMD) regulate virulence traits in both *C. neoformans* and *C. deneoformans*. (**A**) Immunoblot analysis of H2Bub1 and H3K4me in strains expressing the indicated proteins in *C. neoformans* strain background. (**B**) Thermal tolerance and melanin production in *C. neoformans* strain background under indicated conditions. (**C**) Capsule production in the indicated *C. neoformans* strains on capsule-inducing media. The non-capsule-producing strain *nrg1Δ* was used as control. (**D**) The indicated strains in *C. neoformans* background were cultured on capsule-inducing media. Cell body diameter of 50 cells were quantified with ImageJ. (**E**) The ratio between diameters of capsule layer and cell body of the indicated strains in *C. neoformans* background were quantified. 50 cells for each strain were analyzed. (**F**) Melanin production of indicated strains in *C. deneoformans* background in liquid L-DOPA media were quantified by determining $OD_{490}$. Bars show the mean ± SD of three biological replicates. (**G**) Capsule production in the indicated *C. deneoformans* strains on capsule-inducing media. (**H**) The ratio between diameters of capsule layer and cell body of the indicated strains in *C. deneoformans* background was quantified. Fifty cells for each strain were analyzed. Ordinary one-way ANOVA were used to compare differences between groups. ns,not significant; *, $p < 0.05$; **, $p < 0.01$; ****, $p < 0.0001$.

The online version of this article includes the following source data for figure 4:

**Source data 1.** Original files for the western blot analysis displayed in *Figure 4A*.

**Source data 2.** PPT files of western blots for *Figure 4A*, indicating the relevant bands.

We further investigated the effects of Rtf1 on virulence traits in *C. deneoformans*. Given the poor growth of serotype D strains under higher temperature, we only determined the production of melanin and capsule in *C. deneoformans* strains. Consistent with what we observed in *C. neoformans* H99 strain background, the *rtf1Δ* mutant produced less melanin in L-DOPA media, and overexpression of the full-length Rtf1 restored the melanin production and the HMD domain failed to do so (*Figure 4F*). In addition, the cell size and ratio between diameters of capsule layer and cell body in *rtf1Δ* were significantly smaller than the those in the XL280 reference strain (*Figure 4G, H*). Overexpression of the full-length Rtf1 partially restored the ratio to the level as in the XL280 strain, while overexpression the HMD, failed to do so (*Figure 4H*). Surprisingly, overexpression of the Plus3 domain, Rtf1$^{E95A}$ or

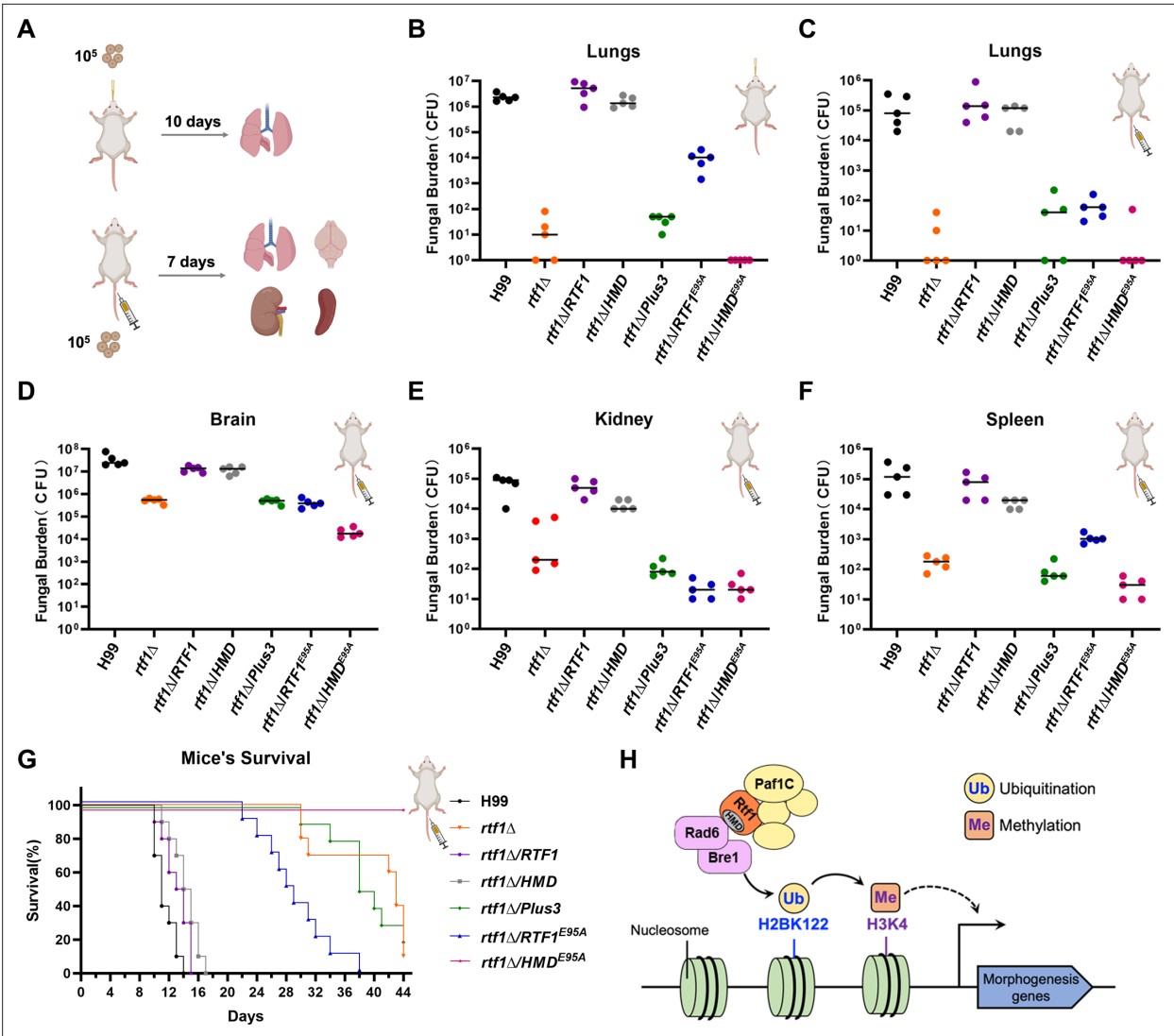

**Figure 5.** Rtf1 and histone modification domain (HMD) regulate cryptococcal pathogenicity in murine models of cryptococcosis. (**A**) Schematic diagram of the intranasal and intravenous infection models of cryptococcosis. The inoculum and time for detecting fungal burden in these two infection models were indicated, respectively. (**B**) The fungal burden of lungs infected by indicated strains through intranasal infection (*n* = 5). (**C–F**) The fungal burden of lungs, brain, kidney, and spleen infected by indicated strains through intravenous infection (*n* = 5). (**G**) The survival curve of animals infected by indicated strains through intravenous infection. The inoculum was the same as the fungal burden assay for intravenous infection. (**H**) Schematic diagram of the working model depicting the role of Rtf1 and its HMD domain in regulating fungal morphogenesis in *C. deneoformans*.

HMD$^{E95A}$ partially restored the melanin production and the ratio between diameters of capsule layer and cell body in the *rtf1Δ* mutant in *C. deneoformans* (*Figure 4F, H*). These results indicated that Rtf1 protein may have serotype-specific regulatory mode-of-actions in *C. neoformans* and *C. deneoformans* regarding to melanin and capsule production.

## Roles of the HMD-mediated H2Bub1 in regulating cryptococcal pathogenicity

To further investigate the role of HMD-mediated H2Bub1 in the pathogenicity of *C. neoformans*, we tested the fungal burdens and survival rates of wild-type, *rtf1Δ*, and complemented strains in intranasal and intravenous murine models of cryptococcosis (*Figure 5A*). Our results showed that both intranasally and intravenously infected lungs by the *rtf1Δ* mutant had significantly reduced fungal burden compared to lungs infected by wild-type, full-length *RTF1*-complemented, or HMD-complemented strains (*Figure 5B, C*). The lungs infected by Plus3-, Rtf1$^{E95A}$-, or HMD$^{E95A}$-complemented strain had

comparable fungal burden relative to the *rtf1Δ*-infected lungs (*Figure 5B, C*). Similar trends in effects on fungal burden were observed in other intravenously infected organs, including brain, kidney, and spleen (*Figure 5D–F*). In consistent with the fungal burden analysis, the pathogenicity of *rtf1Δ* mutant in the intravenous model of cryptococcosis were significantly attenuated compared to the wild-type strain and strains complemented with the full length of Rtf1 or the HMD domain, while the Plus3, Rtf1[E95A], or HMD [E95A] failed to complement the attenuated virulence of the *rtf1Δ* mutant (*Figure 5G*). Together, our findings suggest that HMD-mediated H2Bub1 is essential for the successful survival and proliferation of *C. neoformans* during infection.

## Discussion

In this study, we investigated the role of Rtf1 in promoting H2Bub1 and consequently regulating cryptococcus yeast-to-hypha transition and virulence. Here, we demonstrated that the global H2Bub1 plays pleiotropic roles in the sexual reproduction, morphogenesis, melanin production, thermal tolerance, and pathogenicity of *C. deneoformans* and *C. neoformans*. Interestingly, the Rtf1 HMD domain alone is sufficient to facilitate global H2Bub1 and subsequent H3K4me. The HMD domain could fully restore the deficiencies on filamentation in vitro and pathogenicity in a murine model of cryptococcosis. Our results fit a model in which Rtf1 facilitates the global H2Bub1 and subsequent H3K4me levels, in order to promote expression of genes involved in morphogenesis and pathogenicity in *C. neoformans*.

Paf1C was first identified as the RNA polymerase II transcriptional regulator functioning in transcription elongation, and is also required for Rad6/Bre1-mediated H2Bub1 and subsequent H3K4me. Whether and how these two roles interplay with each other remain unclear. In yeast, Paf1C contains five highly conserved core subunits Paf1, Leo1, Ctr9, Cdc73, and Rtf1, which is stably associated with the other subunits within Paf1C. However, the human core Paf1C was shown to interact with RNA polymerase II, in the absence of human Rtf1 homolog, indicating the dispensable role of human Rtf1 in the function of Paf1C (*Vos et al., 2018*). The Paf1C is conserved and consists of five subunits in *C. neoformans* (*Liu et al., 2023*). To investigate the association of Rtf1 with Paf1C in *C. neoformans*, we conducted co-immunoprecipitation coupled with mass spectrometry (CoIP/MS) assays. None of the other four Paf1C subunits could be detected with either FLAG-tagged full length of Rtf1, HMD, or Plus3 as bait (*Supplementary file 2*), strongly indicating that Rtf1 is not stably associated with the Paf1C in *C. neoformans*.

Rtf1 is critical for H2Bub1 levels, and its deletion abolishes global H2Bub1 in both yeast and humans. It is reasoned that Rtf1 may play dual roles in regulating elongation of gene transcription and deposition of H2Bub1. To gain a comprehensive insight into the function of Rtf1 in eukaryotes, we investigated the role of Rtf1 in human fungal pathogens *C. deneoformans* and *C. neoformans* that belong to Basidiomycota. Besides its conserved functions in facilitating global H2Bub1, we also found it is required for fungal morphogenesis and pathogenicity. We showed that HMD domain alone is sufficient to restore cryptococcal filamentation and virulence (*Figures 1, 3, and 5*), concomitant with the restoration of global H2Bub1 levels (*Figures 1, 3, and 4*). Given that HMD domain alone lacks regions of full-length Rtf1 required for its interactions with other Paf1C subunits and transcribed regions of genes (*Piro et al., 2012*; *de Jong et al., 2008*), our results on the HMD domain support for a model in which the function of Rtf1 in regulating H2Bub1 is uncoupled from interaction with other subunits of Paf1C, and it is required for cryptococcal development and virulence (*Figure 5H*). Biochemical and biophysical studies on the association of Rtf1 with Paf1C subunits and Rad6/Bre1 would provide further insights into its mode-of-action in regulating establishment and deposition of H2Bub1.

Rtf1 contains two conserved domains Plus3 and HMD (*Figure 1E*). The Plus3 domain has been shown to interact with single-stranded DNA, indicating a role for Rtf1 in the elongation bubble during transcription elongation (*de Jong et al., 2008*). In addition, Plus3 may also function in facilitating proper positioning of H2Bub1, especially in regions of actively transcribed genes (*Piro et al., 2012*). Here, we showed that the Plus3 domain alone has no effects on global H2Bub1, while the HMD domain alone could facilitate H2Bub1 (*Figure 1A*). Moreover, the *rtf1Δ* mutant showed reduced thermal tolerance with growth defects at 37 and 39°C, compared to the wild-type strain (*Figure 4C*). The full length of Rtf1 or HMD domain alone fully restored the growth defect of *rtf1Δ* mutant at 37°C; However, both of them only partially restored the growth defect at 39°C (*Figure 4C*). In addition, only the full length

of Rtf1 restored melanin production in the *rtf1Δ* mutant (**Figure 4D**), although the HMD domain alone fully restored the global H2Bub1 levels in *C. neoformans* (**Figure 4A**). There are two possibilities that may lead to these observations: (1) H2Bub1 was not properly depositioned with expression of only HMD domain, although the global level of H2Bub1 seems normal; (2) production of the virulence factors may require functions of Rtf1 and/or Paf1C in transcription elongation, which is absent in HMD-complemented strain (**Piro et al., 2012**; **Van Oss et al., 2016**). These results on the HMD domain in regulating virulence factors provide insights into the function of full-length Rtf1 and interactions with other subunits of Paf1C. A detailed comparison of H2Bub1 occupancies across chromosomes between cells expressing the full length of Rtf1 and HMD alone would be of great interest. In addition, over-expression of truncated version of Rtf1 in serotype A and D background *rtf1Δ* mutants give different output in terms of melanin and capsule production. These unexpected findings indicate that the Rtf1 protein may have different regulatory function in serotype A and D strains, in addition to the shared function in mediating H2Bub1. Further studies are required to uncover the roles of Paf1C and Rtf1 in facilitating proper deposition of H2Bub1 to regulate fungal morphogenesis and pathogenicity.

## Materials and methods

### Strains, culture conditions, and microscopy examination

Strains used in this study are listed in **Supplementary file 1**. *C. deneoformans* and *C. neoformans* strains were maintained on YPD medium unless specified otherwise. Transformants obtained from transient CRISPR–Cas9 coupled with electroporation (TRACE) were selected on YPD medium with 100 µg/ml of nourseothricin, 100 µg/ml of G418, or 200 µg/ml of hygromycin.

Strains for phenotypic assays were grown overnight in liquid YPD medium at 30°C with shaking. The cells were washed with sterile water, adjusted to an optical density at 600 nm ($OD_{600}$) of 3.0, and serially diluted. For filamentation tests, aliquots (3 µl) of cell suspensions ($OD_{600}$ = 3.0) were spotted onto V8 plates and cultured at room temperature in the dark. For morphological examinations, all strains were examined under a stereoscope. For spotting assays, aliquots (3 µl) of serial dilutions starting from $OD_{600}$ = 3.0 were spotted onto agar medium with supplements and cultured under the noted conditions.

### Gene manipulation

Cryptococcal genes were deleted following the TRACE protocol (**Fan and Lin, 2018**; **Lin et al., 2020**). In brief, a deletion construct with approximately 1 kb of homologous arms flanking a target gene and the dominant marker was cloned through fusion PCR. This construct was mixed with PCR products of *CAS9* and a relevant guide RNA (gRNA), and the mixture was introduced into recipient strains by electroporation as described previously (**Lin et al., 2020**). Resulting yeast colonies were screened by two rounds of diagnostic PCR. The first round of PCR was performed to detect the integration of the construct into the corresponding locus of the target gene. The second round of PCR was performed to confirm knockout of the target fragment. All primers used to make gene deletion mutants are listed in **Supplementary file 1**.

For gene complementation, the ORFs plus approximately 1.0 kb of their upstream regions were amplified by PCR and cloned into vectors through T5 exonuclease-dependent assembly as previously described (**Xia et al., 2019**). For gene overexpression with inducible or constitutively active promoters, the constructs were obtained by amplifying the entire ORF by PCR and cloning the PCR products into vectors at the downstream of *CTR4*, *TEF1*, or *GPD1* promoter. All plasmids were confirmed by restriction enzyme digestion and sequencing. The confirmed constructs, together with PCR products of *CAS9* and gRNA targeting the Safe Haven locus (**Fan and Lin, 2020**; **Upadhya et al., 2017**), were introduced into recipient *Cryptococcus* strains. The transformants were passaged once per day for 5 days and cultured on selection plates to obtain stable transformants. Then, two rounds of diagnostic PCR were performed to confirm the integration and orientation of constructs into the Safe Haven locus. All primers and plasmids used for gene complementation and overexpression are listed in **Supplementary file 1**.

### Protein extraction and western blotting

Proteins were extracted from *Cryptococcus* cells according to a previously described method (**Zhao and Lin, 2021**). Aliquots of proteins were separated on 4–12% gradient SDS–PAGE gels and then

transferred to a polyvinylidene difluoride membrane for western blot analyses. Antibodies used in this study are listed in *Supplementary file 1*. For CoIP/MS, whole-cell extracts of experimental strains were incubated with FLAG-trap (Sigma) according to the manufacturer's instructions. Proteins in the eluted samples were loaded in SDS–PAGE gel, digested, and analyzed by the Proteome Facility Centre of Institute of Microbiology, Chinese Academy of Sciences.

## RNA extraction and qPCR assays

*Cryptococcus* strains were cultured in liquid YPD with shaking at 220 rpm at 30°C overnight, or on solid V8 medium at room temperature in the dark for 24 hr. The cultures were collected, flash frozen in liquid nitrogen, and lyophilized for 24 hr. Total RNA was isolated with the PureLink RNA Mini Kit (Invitrogen), and first strand cDNA was synthesized using the GoScript Reverse Transcription System (Promega) following the manufacturer's instructions. The Power SYBR Green system (Invitrogen) was used for RT-PCR. All the primers used here are listed in *Supplementary file 1*. Relative transcript levels were determined using the ΔΔCt method as described previously. Three biological replicates were included for all tests. Statistical significance was determined using a Student's *t*-test. Differences for which $p < 0.05$ were considered statistically significant.

## RNA-seq and data analysis

For RNA-seq analyses, strains were cultured in YPD liquid medium at 30°C overnight. The cells were washed with $ddH_2O$ and spotted on V8 medium to stimulate unisexual reproduction. The level and integrity of RNA in each sample were evaluated using a Qubit RNA Assay Kit on a Qubit 2.0 Fluorometer (Life Technologies, CA, USA) and RNA Nano 6000 Assay Kit with the Bioanalyzer 2100 system (Agilent Technologies, CA, USA), respectively. RNA purity was assessed using a Nano Photometer spectrophotometer (IMPLEN, CA, USA). The transcriptome libraries were generated using the VAHTS mRNA-seq v2 Library Prep Kit (Vazyme Biotech Co., Ltd, Nanjing, China) according to the manufacturer's instructions.

The transcriptome libraries were sequenced by Annoroad Gene Technology Co., Ltd (Beijing, China) on an Illumina platform. For RNA-seq analysis, the quality of sequenced clean data was analyzed using FastQC software. Subsequently, sequences from approximately 2 GB of clean data for each sample were mapped to the genome sequence of *C. deneoformans* XL280α using STAR. Gene expression levels were measured in transcripts per million by Stringtie to determine unigenes. All unigenes were subsequently aligned against the well-annotated genome of JEC21, which served as the parent strain to generate XL280α through a cross with B3501α. The differential expression of genes was assessed using DEseq2 of the R package and defined based on the fold-change criterion ($\log_2$|fold-change|>1.0, adjusted p-value <0.05).

## Virulence trait assays

Strains for examining virulence factors were grown overnight in liquid YPD at 30°C with shaking. The overnight cultures were washed with sterile water, adjusted to $OD_{600} = 3.0$, and serially diluted. For thermal tolerance, melanin production, and capsule formation assay on solid plates, aliquots (3 µl) of serially diluted cell suspensions were spotted onto YPD plate, L-dopamine media, and 10% FBS media (*Vartivarian et al., 1993*), respectively. Thermal tolerance was test at 30, 37, and 39°C; melanin production was tested at 30°C in the dark; capsule formation was tested at 37°C with 5% $CO_2$ for 48 hr. The capsule was visualized by staining with India ink and observed under a microscope. Images were captured using Zeiss Axio Imager M2 microscope. At least 50 cells were quantified and processed in ImageJ software to measure capsule and body sizes. All assays were repeated at least three times.

For melanin production assay in liquid media, the overnight cultures were washed with sterile water, adjusted to $OD_{600} = 0.3$ and transferred to L-DOPA medium (containing 0.1% L-asparagine, 0.1% dextrose, 3 mg/ml $KH_2PO_4$, 0.25 mg/ml thiamine, 5 ng/ml biotin, 0.2 mg/ml $MgSO_4 \cdot 7H_2O$ and 1 µg/ml L-DOPA). H99 strains grown for 48 hr at 30°C, each strain counted and diluted at $2 \times 10^7$ cells/ml, while XL280 cells was induced at 30°C for 96 hr and diluted at $7 \times 10^7$ cells/ml. $OD_{490}$ was measured to determine melanin production.

## Murine models of cryptococcosis

### Intranasal infection model

Female Balb/C mice of 8–10 weeks old were purchased from the Laboratory Animal Center of Zhengzhou University, China. Cryptococcal strains were inoculated in 3 ml of liquid YPD medium with the initial $OD_{600}$ = 0.2 (approximately $10^6$ cell/ml) and incubated for 15 hr at 30°C with shaking. Prior to intranasal infection, cells were washed with sterile saline three times and adjusted to the final concentration of $2 \times 10^6$ cell/ml. Once the mice were sedated with ketamine and xylazine via intraperitoneal injection, 50 µl of the cell suspension ($1 \times 10^5$ cells per mouse) were inoculated intranasally as previously described (*Liu et al., 2023*; *Lin et al., 2022*; *Zhao et al., 2020*; *Zhai et al., 2013*). Mice were monitored daily for disease progression. Animals were euthanized at 10 DPI, and lungs were dissected for fungal burden quantification.

### Intravenous infection model

Prior to intravenous infections, cryptococcal cells were washed with sterile saline three times and adjusted to the final concentration of $2 \times 10^6$ cell/ml. Mice were sedated with Isoflurane. 50 µl of the cell suspension ($1 \times 10^5$ cells per mouse) were injected intravenously as previously described (*Liu et al., 2023*; *Lin et al., 2022*; *Zhao et al., 2020*; *Zhai et al., 2013*). After DPI 7, animals were euthanized, and the brain, lungs, kidneys, and spleens were dissected.

For fungal burden quantifications, dissected organs were homogenized in 2 ml of cold sterile PBS. Tissue suspensions were serially diluted in PBS and plated onto YNB agar medium and incubated at 30°C for 2 days before counting the CFUs.

## DAPI staining

DAPI (4',6-diamidino-2-phenylindole) staining assays were performed as previously described (*Zhao et al., 2018*). Briefly, yeast cells or hyphae were collected and fixed with 3.7% formaldehyde and permeabilized in 1% Triton X-100. The cells were then washed three times with PBS and incubated in 2 µg/ml DAPI before being dropped onto a glass slide for fluorescent microscopic observation.

## Acknowledgements

We thank the Zhao lab for their continued interest and ideas. We thank the Big Data Center and Bioinformatics Center at Department of Veterinary Medicine, Henan Agricultural University for providing high-performance computing service. This work was supported by National Natural Science Foundation of China (nos. 32373093 and 30900880 to Zhao Y; 32402947 to Zhang J) and Henan Agricultural University (no. 30500946 to Zhao Y).

## Additional information

### Funding

| Funder | Grant reference number | Author |
| --- | --- | --- |
| National Natural Science Foundation of China | 32373093 | Youbao Zhao |
| National Natural Science Foundation of China | 32402947 | Jing Zhang |
| Henan Agricultural University | 30500946 | Youbao Zhao |
| National Natural Science Foundation of China | 30900880 | Youbao Zhao |

The funders had no role in study design, data collection, and interpretation, or the decision to submit the work for publication.

## Author contributions

Yixuan Jiang, Data curation, Formal analysis, Validation, Investigation, Visualization, Writing – original draft; Ying Liang, Data curation, Formal analysis, Validation, Visualization, Writing – review and editing; Fujie Zhao, Zhenguo Lu, Siyu Wang, Yao Meng, Zhanxiang Liu, Data curation, Investigation; Jing Zhang, Conceptualization, Resources, Data curation, Formal analysis, Supervision, Funding acquisition, Writing – original draft, Project administration, Writing – review and editing; Youbao Zhao, Conceptualization, Resources, Data curation, Software, Formal analysis, Supervision, Funding acquisition, Validation, Investigation, Visualization, Methodology, Writing – original draft, Project administration, Writing – review and editing

## Author ORCIDs

Youbao Zhao https://orcid.org/0000-0003-2406-3922

## Ethics

All animal experiments were reviewed and ethically approved by the Committee on the Ethics of Animal Care and Use of National Research Center for Veterinary Medicine (20210422074) and were carried out in accordance with the regulations in the Guide for the Care and Use of Animals in Research of the People's Republic of China. Infections with C. neoformans were performed via the intranasal route. Eight- to ten-week-old female Balb/c mice were purchased from the Center of Experimental Animal of Zhengzhou University (Zhengzhou, China) and used for survival and fungal burden analyses.

Reviewer #1 (Public review): https://doi.org/10.7554/eLife.99229.3.sa1
Author response https://doi.org/10.7554/eLife.99229.3.sa2

---

# Additional files

## Supplementary files

Supplementary file 1. Strains, plasmids, primers, and antibodies used in this study.

Supplementary file 2. Potential interacting proteins with Rtf1, histone modification domain (HMD), and Plus3 identified by co-immunoprecipitation coupled with mass spectrometry (CoIP/MS).

MDAR checklist

## Data availability

RNA sequencing data have been deposited in GEO under accession number GSE296031.

The following dataset was generated:

| Author(s) | Year | Dataset title | Dataset URL | Database and Identifier |
|---|---|---|---|---|
| Jiang Y, Liang Y, Zhang J, Zhao Y | 2025 | Rtf1 HMD domain facilitates global histone H2B monoubiquitination and regulates morphogenesis and virulence in the meningitis-causing pathogen Cryptococcus neoformans | https://www.ncbi.nlm.nih.gov/geo/query/acc.cgi?acc=GSE296031 | NCBI Gene Expression Omnibus, GSE296031 |

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
