## [Editor Report · eLife Assessment]

This is an **important** study that connects the polymerase-associated factor 1 complex (Paf1C) with Histone 2B monoubiquitination and the expression of genes key to virulence in Cryptococcus neoformans. The provided information is **convincing** and has the potential to open several opportunities to further understand the basic biology of this significant human fungal pathogen.

---

## [Referee Report · Reviewer #1 (Public review)]

In the manuscript entitled "Rtf1 HMD domain facilitates global histone H2B monoubiquitination and regulates morphogenesis and virulence in the meningitis-causing pathogen Cryptococcus neoformans" by Jiang et al., the authors employ a combination of molecular genetics and biochemical approaches, along with phenotypic evaluations and animal models, to identify the conserved subunit of the Paf1 complex (Paf1C), Rtf1, and functionally characterize its critical roles in mediating H2B monoubiquitination (H2Bub1) and the consequent regulation of gene expression, fungal development, and virulence traits in C. deneoformans or C. neoformans. Specially, the authors found that the histone modification domain (HMD) of Rtf1 is sufficient to promote H2B monoubiquitination (H2Bub1) and the expression of genes related to fungal mating and filamentation, and restores the fungal morphogenesis and pathogenicity defects caused by RTF1 deletion. These findings highlight the critical contribution of Rtf1's HMD to epigenetic regulation and cryptococcal virulence. This work will be of interest to fungal biologists and medical mycologists, particularly those studying fungal epigenetic regulation and fungal morphogenesis.

Comments on revisions:

The revised manuscript addresses all my previous concerns satisfactorily.

---

## [Author Response]

The following is the authors’ response to the original reviews

**Public Reviews:**

**Reviewer #1 (Public Review):**
Summary:In the manuscript entitled "Rtf1 HMD domain facilitates global histone H2B monoubiquitination and regulates morphogenesis and virulence in the meningitis-causing pathogen Cryptococcus neoformans" by Jiang et al., the authors employ a combination of molecular genetics and biochemical approaches, along with phenotypic evaluations and animal models, to identify the conserved subunit of the Paf1 complex (Paf1C), Rtf1, and functionally characterize its critical roles in mediating H2B monoubiquitination (H2Bub1) and the consequent regulation of gene expression, fungal development, and virulence traits in C. deneoformans or C. neoformans. Specially, the authors found that the histone modification domain (HMD) of Rtf1 is sufficient to promote H2B monoubiquitination (H2Bub1) and the expression of genes related to fungal mating and filamentation, and restores the fungal morphogenesis and pathogenicity defects caused by RTF1 deletion.Strengths:The manuscript is well-written and presents the findings in a clear manner. The findings are interesting and contribute to a better understanding of Rtf1-mediated epigenetic regulation of fungal morphogenesis and pathogenicity in a major human fungal pathogen, and potentially in other fungal species, as well.Weaknesses:A major limitation of this study is the absence of genome-wide information on Rtf1-mediated H2B monoubiquitination (H2Bub1), as well as a lack of detail regarding the function of the Plus3 domain. Although overexpression of HMD in the rtf1Δ mutant restored global H2Bub1 levels, it did not rescue certain critical biological functions, such as growth at 39 °C and melanin production (Figure 4C-D). This suggests that the precise positioning of H2Bub1 is essential for Rtf1's function. A comprehensive epigenetic landscape of H2Bub1 in the presence of HMD or full-length Rtf1 would elucidate potential mechanisms and shed light on the function of the Plus3 domain.

We thank the reviewer (and other reviewers) for this excellent suggestion. We have conducted CUT&Tag assays with WT, *rtf1*Δ mutant, and complementary strains with the full length Rtf1 and only HMD domain cultured under 30 and 39 °C. We indeed found that the epigenetic landscape of H2Bub1 in the presence of HMD or full-length Rtf1 has variations. This results strongly suggest that the distribution of H2Bub1 is regulated by Rtf1, and H2B modifications at specific loci in the chromosome may contribute to thermal tolerance in C. neoformans. These new findings from CUT&Tag assays shed lights on understanding the mechanism of thermal tolerance, and we decided not to include these results in the current manuscript.

**Reviewer #2 (Public Review):**
Summary:The authors set out to determine the role of Rtf1 in Cryptococcal biology, and demonstrate that Rtf1 acts independently of the Paf1 complex to exert regulation of Histone H2B monoubiquitylation (H2Bub1). The biological impact of the loss of H2Bub1 was observed in defects in morphogenesis, reduced production of virulence factors, and reduced pathogenic potential in animal models of cryptococcal infection.Strengths:The molecular data is quite compelling, demonstrating that the Rtf1-depednent functions require only this histone modifying domain of Rtf1, and are dependent on nuclear localization. A specific point mutation in a residue conserved with the Rtf1 protein in the model yeast demonstrates the conservation of that residue in H2Bub1 modification. Interestingly, whereas expression of the HMD alone suppressed the virulence defect of the rtf1 deletion mutant, it did not suppress defects in virulence factor production.Weaknesses:The authors use two different species of Cryptococcus to investigate the biological effect of Rtf1 deletion. The work on morphogenesis utilized C. deneoformans, which is well-known to be a robust mating strain. The virulence work was performed in the C. neoformans H99 background, which is a highly pathogenic isolate. The study would be more complete if each of these processes were assessed in the other strain to understand if these biological effects are conserved across the two species of Cryptococcus. H99 is not as robust in morphogenesis, but reproducible results assessing mating and filamentation in this strain have been performed. Similarly, C. deneoformans does produce capsule and melanin.

We thank the reviewer for the suggestion. We have conducted assays to quantify both capsule and melanin production in both C. neoformans and C. deneoformans strain background. We found that capsule production was affected in the same pattern in these two serotypes. Interestingly, we found the cell size was significantly affected by deletion of RTF1 in both serotypes. In addition, melanin production was reduced due to the deletion of RTF1 in both serotypes; However, complementation with Plus3 or mutated alleles of HMD gave different phenotypes in these two serotypes. These new findings were included Figure 4 in the revised manuscript.

There are some concerns with the conclusions related to capsule induction. The images reported in Figure B are purported to be grown under capsule-inducing conditions, yet the H99 panel is not representative of the induced capsule for this strain. Given the lack of a baseline of induction, it is difficult to determine if any of the strains may be defective in capsule induction. Quantification of a population of cells with replicates will also help to visualize the capsular diversity in each strain population.

We thank the reviewer for raising this concern. We have tested capsule production under capsule-inducing condition on 10% fetal bovine serum (FBS) agar medium [1]. Under this condition, the capsule layers surrounding the cells were obvious. We also included noncapsule-producing control in our assay to help the visualization of capsule. In addition, we quantified the ratio between diameters of capsule layer and cell body to show the capsular diversity in each strain population. The results were included in the Figure 4 in the revised manuscript.

The authors demonstrate that for specific mating-related genes, the expression of the HMD recapitulated the wild-type expression pattern. The RNA-seq experiments were performed under mating conditions, suggesting specificity under this condition. The authors raise the point in the discussion that there may be differences in Rtf1 deposition on chromatin in H99, and under conditions of pathogenesis. The data that overexpression of HMD restores H2Bub1 by western is quite compelling, but does not address at which promoters H2Bub1 is modulating expression under pathogenesis conditions, and when full-length Rtf1 is present vs. only the HMD.

We thank the reviewer for raising these concerns. Please see our response to Reviewer #1.

**Reviewer #3 (Public Review):**
Summary:In this very comprehensive study, the authors examine the effects of deletion and mutation of the Paf1C protein Rtf1 gene on chromatin structure, filamentation, and virulence in Cryptococcus.Strengths:The experiments are well presented and the interpretation of the data is convincing.Weaknesses:Yet, one can be frustrated by the lack of experiments that attempt to directly correlate the change in chromatin structure with the expression of a particular gene and the observed phenotype. For example, the authors observed a strong defect in the expression of ZNF2, a known regulator of filamentation, mating, and virulence, in the rtf1 mutant. Can this defect explain the observed phenotypes associated with the RTF1 mutation? Is the observed defect in melanin production associated with altered expression of laccase genes and altered chromatin structure at this locus?

We completely agree with the reviewer. We have conducted CUT&Tag assay, and checked the Rtf1-mediated H2Bub1 at these particular gene loci. We found that the distribution of H2Bub1 at the promoter region of ZNF2 and the gene body of laccase-encoding gene varied possibly due to RTF1 mutation. We would like to save those preliminary findings for another story and not to include in this manuscript as we mentioned in the response to Reviewer #1.

(1) Jang, E.-H., et al., Unraveling Capsule Biosynthesis and Signaling Networks in Cryptococcus neoformans. Microbiology Spectrum, 2022. 10(6): p. e02866-22.